# PROMPT PROGRAMMING FOR THE VISUAL DOMAIN

**Alayt Issak**
Northeastern University
Boston, MA 02120, USA
`issak.a@northeastern.edu`

**Lav R. Varshney**
University of Illinois Urbana-Champaign
Urbana, Illinois 61801, USA
`varshney@illinois.edu`

## ABSTRACT

In this work, we ask how text-to-image synthesis via large language models can effectively probe imagery that embodies fidelity and imagination. We investigate this question in the context of prompts (writing to language models) in a novel probing mechanism known as prompt programming, or programming in natural language. We start by refining existing techniques to characterize the effect of templates on visual fidelity then hone in on approaches to capture holistic nuances within the visual domain. We present a systematic analysis of prompt engineering for visual image generation.

## 1 INTRODUCTION

AI models for text-prompted visual image generation have become very prominent in recent months PromptBase (2022) and we find it important to determine how to use these models so they may become useful tools for expressing creative intentionality. As such, we consider prompt programming to probe a model's capabilities in light of their specific benefit to language models, as quantified in recent work to often worth 100s of data points in enhancing a model's ability Scao & Rush (2021). We find prompts to be a valuable method for invoking the development of language models' novel capabilities and transfigure their benefits towards advancements in the visual domain.

## 2 NATURAL LANGUAGE TEMPLATES

We begin by referencing six templates from already established natural language prompt programming templates Reynolds & McDonell (2021). From there, we systematically analyze and narrow down to three templates, Metaprompt techniques along with a Memetic proxy and Constraining behavior, as they yield promising results (Appendix A). A description of the chosen templates is provided below:

1. **Memetic proxy:** task specification by prompting alongside a proxy to signify intention using memetic reinforcement techniques such as an archetypal situation.

2. **Constraining behavior:** specified prompting to enforce desired behavior by incorporating emphasis such as punctuation marks.

3. **Metaprompt programming:** prompting seeds encapsulating a more general intention that will unfold into a specific prompt along with additional information such as a prompt wrapper for the task to be solved.

## 3 LEARNED METAPROMPT TECHNIQUES

To generalize a framework specific to the visual domain, we begin by defining four Metaprompts that encapsulate a prompt as a separate entity. These are **L-Meta:** (Let's ...), **T-Meta:** (This prompt asks us to...), **I-Meta:** (In this prompt, we...), and **S-Meta:** (Shall we draw...). We then conceptualize our technique on a variety of concepts by devising inhabitants of a modern, botanical farm as modes of exploration. We chose this mode as they are generalizations of concepts. Our inhabitants include tomatoes, wheat, a robot, and fungi. Next, we set an exclamation mark as our constraining

behavior for imperative imagery and Unreal Engine (a popular gaming software known for its pixelated computer graphics) as our memetic proxy. As an inanimate proxy, Unreal Engine overpasses the model's inclination to infer a specified entity within the domain of interest and instead infers to the stylistic features of the methodology. Finally, we present our images generated via the CLIP + VGQAN model Crowson et al. (2022) in Figure 1 below:

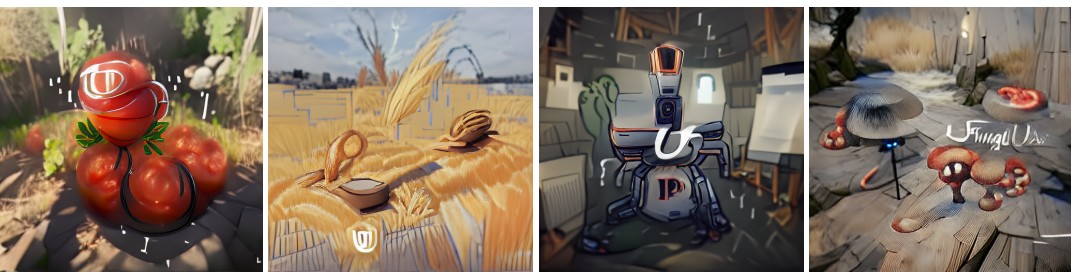

Figure 1: | **L-Meta:** *Let's draw tomatoes in the style of an unreal engine!* | **T-Meta** *This prompt asks us to draw wheat in the style of an unreal engine!* | **I-Meta:** *In this prompt, we draw a robot in the style of an Unreal Engine 5!* | **S-Meta:** *Shall we draw fungi in the style of Unreal Engine 5!*

## 4 BEYOND TEMPLATES

Metaprompts yield qualitative results yet signify the need for an alternate behavior. For example, while each image in Figure 1 was roughly trained in 1000 iterations, some were chosen interactively during various time stamps in part of the imagery that arose during training. Likewise, given that the resultant imagery is also subject to creativity via the prompt's intention or the prompter's interpretation, we infer descriptive prompts that can themselves probe the capability of a model towards stark fidelity. We illuminate this ideation in two detailed formats on the subject of electrical activity in fungi via creative prompt programming techniques in Figure 2 below.

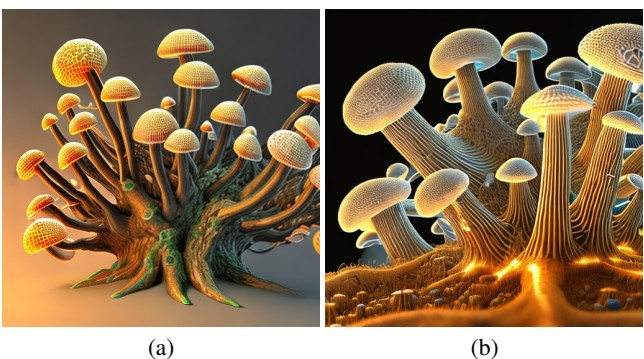

(a)         (b)

Figure 2: (a) An intricate sci-fi VR 3D painting of electrical activity in Fungi showing the spiking activity of the mycelium networks with movement about mechanisms and (b) An intricate sci-fi VR 3D painting of electrical activity in Fungi showing the spiking activity of the mycelium networks detecting the activity about mechanisms.

## 5 CONCLUSION

We present this study for the scholarly and artistic grounding of prompt programming within the visual domain. As such, we encourage practitioners to emphasize the holistic notion of text-to-image synthesis and explore creative prompting techniques alongside Metaprompts and prescriptive templates. As practicing artists, we also put forth these findings with the underlining ethical care to safeguard the curation and deployment of such generated artifacts Fuchs et al. (2020); Issak & Varshney (2022).

URM STATEMENT

The authors acknowledge that at least one key author of this work meets the URM criteria of ICLR 2023 Tiny Papers Track.

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

## A   APPENDIX

### A.1   NATURAL LANGUAGE TEMPLATES (NLT)

In the following analysis, we use the base prompt of *Draw a boy* to benchmark across the six templates provided by Reynolds and McDonell Reynolds & McDonell (2021). Images are generated via the original CLIP + VQGAN notebook Crowson et al. (2022) for up to 500 iterations and do not use an initial or target image to guide the generation with inklings to desires.

| NLT 1 | Image | Characteristic |
|---|---|---|
| **Direct task specification:** a zero-shot prompt which tells the model to perform some task by constructing the signifier. | 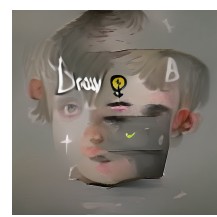 | This figure begins with the initial image of a boy yet incorporates the literal word *Draw* into the image which does not carry the desired implementation. This leads us to specify in the discourse of "How?" to gauge the intended task at hand. |

Table 1: Prompt: *Draw a boy*

| NLT 2 | Image | Characteristic |
|---|---|---|
| **Providing demonstration:** a few-shot prompt to communicate via examples to aid generation in a contextually informative manner. | 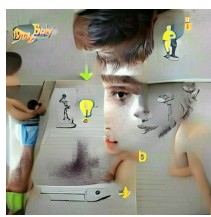 | The figure is scattered and does not carry a unified image of the boy, which could allude to the obscure nature of one's imagination. However, the emphasis of *Draw* is diminished and leads the method as an advocate for task specification. |

Table 2: Prompt: *Draw a boy by using your imagination*

| NLT 3 | Image | Characteristic |
|---|---|---|
| **Memetic proxy:** task specification by prompting alongside a proxy to signify intention using memetic reinforcement techniques such as an archetypal situation. | 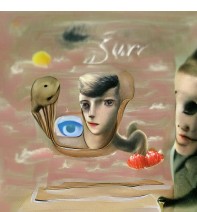 | Although surrealism saw many others such as Joan Miró, this figure has specified to the most renowned archetype of Spanish Surrealist painter Salvador Dalí with the distinct blue eye from his notable piece *The Eye* and landscape which resembles his painting, *The Persistence of Memory*. |

Table 3: Prompt: *Draw a boy like a surrealist*

| NLT 4 | Image | Characteristic |
|---|---|---|
| **Constraining behavior:** specified prompting to enforce desired behavior by incorporating emphasis such as punctuation marks. | 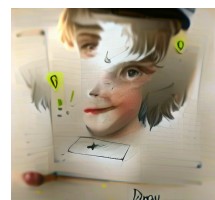 | The constraint applied here via an exclamation mark has gravitated the image to greater clarity as compared to Table 1 which directed focus towards writing *Draw*. We also note that Table 4 has placed the boy on a canvas which alludes to the action of drawing as an artistic pursuit. |

Table 4: Prompt: *Draw a boy!*

| NLT 5 | Image | Characteristic |
|---|---|---|
| **Serializing reasoning:** task specification by prompting via sub-tasks such as step-by-step descriptions that open room for sequential reasoning in the prompt analysis. | 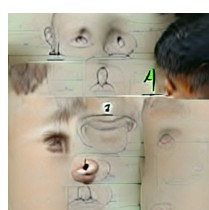 | The method shows similar drawbacks to NLT1 which takes the prompt for its literal meaning. The numbers emerge in response to the outlined sequence and the image of the boy is accordingly split into the corresponding facial features as separate entities. |

Table 5: Prompt: *Draw a boy in the following order of head, eyes, ear, nose, and mouth*

| NLT 6 | Image | Characteristic |
|---|---|---|
| **Metaprompt programming:** entails prompting seeds encapsulating a more general intention that will unfold into a specific prompt when combined with additional information such as a serial explanation of a procedure to solve the problem or a prompt wrapper for the task to be solved. | 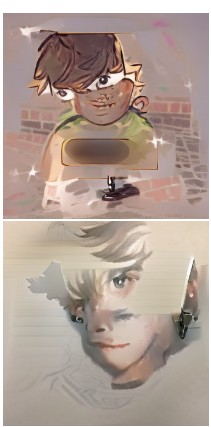 | We implement two seeds: one geared towards an explanation of the procedure of asking the model (Prompt 1) and another that wraps the task on its left-hand side (Prompt 2), respectively. In both cases, the insertion of the prompt seed (*us*) generates images of the greatest clarity. We also witness that a common notion to NLT techniques is to underline meaning that is "read between the lines", which Metaprompts entail. |

Table 6: Prompt 1: *This prompt asks us to draw a boy* and Prompt 2: *Let's draw a boy*

From the exploration of the aforementioned templates, along with their visual fidelity, we notice that Metaprompts eliminate incoherent past instances from previous templates. This includes text from task specifications in Table 1, disentangled generation in Table 2, and an external eye to the image in Table 5. This leads us to carry forth with Metaprompt as a promising approach to derive the base of our natural language template for visual exploration. However, given that certain templates are more successful than others in certain domains, we combine our approaches in forthcoming templates.

