# OpenReview forum: "Prompt Programming for the Visual Domain"
_ICLR.cc/2023/TinyPapers — Submitted to Tiny Papers @ ICLR 2023_

### Official Review · Reviewer_M7h6 · 2023-03-31

**Confidence:** 1

**Summary Of Contributions:**

The authors ask how llm's for text to image synthesis can probe imagery that embodies fidelity and imagination using prompt programming.

**Rating:**

Great Start (GS): a submission which meets some of the reviewing criteria but has room for improvement

**Strengths And Weaknesses:**

Strengths:
- The paper is well-written and clear.

Weakness:
- The technical analysis should be stronger to support the assessments made in section 4.
- Reproducibility could be further improved upon.

**Suggested Changes:**

Changes:
- In page 1, it is mentioned that "We begin our exploration by referencing six templates..". Three of them from (Reynolds & McDonnell) have been mentioned below. It's a bit unclear what the other 3 are?

---

### Official Review · Reviewer_nXdJ · 2023-03-31

**Confidence:** 4

**Summary Of Contributions:**

This paper examines the influence of different kinds of prompts on text to image models. Specifically, they look at how different kinds of prompts affect the visual fidelity of the resulting images.

**Rating:**

Great Start (GS): a submission which meets some of the reviewing criteria but has room for improvement

**Strengths And Weaknesses:**

Strengths:
- Clarity: Relevant related work is well discussed, findings are communicated somewhat clearly

- Follows basic requirements: Paper is within the page limits and complies with formatting requirements

Weaknesses:
- Correctness: Missing details on how the experiments justify and support the claims made

- Reproducibility: Missing some details on how the experiment was done

**Suggested Changes:**

Thank you for your submission! I think the idea of looking at the impact of different prompts on text to image models is very interesting.

Overall I think this paper is a great start, but I felt that it would benefit from some more details surrounding the experimental setup and conclusions of the findings. These additions would allow for better reproducibility of your experiments and the ability to asses the correctness of the claims made in the paper. I think these two points go together, because the better your experiments can be understood, the easier it is to asses the correctness of the findings.

For the details surrounding the experimental setup, it would be great to include more about what your experiment did. For example, how did you choose prompts? How did you then interact with the models you chose once you decided your prompts? Why did you choose tomatoes, wheat, a robot, and fungi? What is the significance of the exclamation point in the prompts?

The conclusion is that creative prompting makes a difference on the visual fidelity of the images generated, but it would be helpful to add some context on how you came to this conclusion and why your experiments support this conclusion. For example, why do the images in Figure 1 tell me that the type of prompt makes a difference? Since each prompt was in a different meta-prompt style but was also asking to draw different objects, it's hard to tell if the meta prompt styles made a difference. Also, why is it enough to draw conclusions from just one image per meta prompt?

---

### Official Review · Reviewer_MvpT · 2023-04-02

**Confidence:** 3

**Summary Of Contributions:**

This paper presents a study on prompt programming for visual image generation

**Rating:**

Needs Clarification (NC): a submission which does not meet the reviewing criteria and needs clarification for its described problem or solution

**Strengths And Weaknesses:**

### Weaknesses ###
- Clarity: The motivation behind this study is not entirely clear, although it points towards a more systematic analysis of prompt engineering for visual image generation.
- Correctness: Given the lack of clear direction in the paper, I am not sure how correct and relevant the results are to the core motivation behind the paper.



**Suggested Changes:**

The authors need to place their work more specifically within the realm of either prompt engineering and describe exactly what novel contribution does their work offer to the field.

---

### Comment · Area_Chair_7rWT · 2023-06-04
**Revised version**

This work meets the threshold for archival, contains the URM statement and is deanonymized.

---

### Meta-Review · Area_Chair_7rWT · 2023-04-08

**Recommendation:** Invite to revise
**Confidence:** 4

**Metareview:**

The influence of different kinds of prompts on text-to-image models is examined.

Written well, but unclear about the direction the authors want the paper to go and how they get there. Missing details on how the experiments justify and support the claims made and hence reproducibility is hard. Authors must use the Appendix to expand and explain the reviewer’s suggestions.


**Summary:**

The influence of different kinds of prompts on text-to-image models is examined.

**Comments And Feedback To The Authors:**

The authors need to do a good revision of the paper. Read the reviewers' comments and suggestions.

**Reason For Not Giving A Higher Recommendation:**

The paper's aim and the experiments' details are unclear. The lack of clarity makes the assessment of correctness and reproducibility hard to analyse.

**Reason For Not Giving A Lower Recommendation:**

N/A

---

### Decision · Program_Chairs · 2023-04-08

Revision accepted; invite to archive